# Current State of Knowledge about Role of Pets in Zoonotic Transmission of SARS-CoV-2

**DOI:** 10.3390/v13061149

**Published:** 2021-06-16

**Authors:** Mateusz Dróżdż, Paweł Krzyżek, Barbara Dudek, Sebastian Makuch, Adriana Janczura, Emil Paluch

**Affiliations:** 1Laboratory of RNA Biochemistry, Institute of Chemistry and Biochemistry, Freie Universität Berlin, Takustraße 6, 14195 Berlin, Germany; 2Department of Microbiology, Wrocław Medical University, St. T. Chałubińskiego 4, 50-376 Wrocław, Poland; pawel.krzyzek@umed.wroc.pl (P.K.); adriana.janczura@umed.wroc.pl (A.J.); 3Laboratory of Microbiology, Private Health Care Institution, St. Jana Pawła II, 41-100 Siemianowice Śląskie, Poland; basia269@op.pl; 4Department of Pathology, Wrocław Medical University, St. K. Marcinkowskiego 1, 50-368 Wrocław, Poland; sebastian.mk21@gmail.com

**Keywords:** SARS-CoV-2, COVID-19, pets suitable animal models, zoonotic potential

## Abstract

Pets play a crucial role in the development of human feelings, social life, and care. However, in the era of the prevailing global pandemic of COVID-19 disease caused by the severe acute respiratory syndrome coronavirus 2 (SARS-CoV-2), many questions addressing the routes of the virus spread and transmission to humans are dramatically emerging. Although cases of SARS-CoV-2 infection have been found in pets including dogs, cats, and ferrets, to date there is no strong evidence for pet-to-human transmission or sustained pet-to-pet transmission of SARS-CoV-2. However, an increasing number of studies reporting detection of SARS-CoV-2 in farmed minks raises suspicion of potential viral transmission from these animals to humans. Furthermore, due to the high susceptibility of cats, ferrets, minks and hamsters to COVID-19 infection under natural and/or experimental conditions, these animals have been extensively explored as animal models to study the SARS-CoV-2 pathogenesis and transmission. In this review, we present the latest reports focusing on SARS-CoV-2 detection, isolation, and characterization in pets. Moreover, based on the current literature, we document studies aiming to broaden the knowledge about pathogenicity and transmissibility of SARS-CoV-2, and the development of viral therapeutics, drugs and vaccines. Lastly, considering the high rate of SARS-CoV-2 evolution and replication, we also suggest routes of protection against the virus.

## 1. Introduction

Due to the prevailing pandemic of the coronavirus disease (COVID-19) caused by severe acute respiratory syndrome coronavirus 2 (SARS-CoV-2), we are forced to observe physical isolation, the reason of which is confirmed human-to-human virus transmission. As of May 2021, a total of 192 countries and territories have documented nearly 170 million confirmed COVID-19 cases and 3.5 million related deaths [1]. The dramatically increasing number of cases adversely affected the global economy, tourism, and the health sectors [2].

The ownership of a companion animal, such as a dog or a cat, is a well-known factor in providing considerable health, social and emotional benefits to their owners. However, at the beginning of COVID-19 pandemic, social anxiety and the psychological repercussions of anecdotal media reports regarding the role of pets in SARS-CoV-2 transmission to humans lead to a dramatic increase in pet abandonment [3]. Nevertheless, over time, reports indicated a low probability of SARS-CoV-2 transmission from pets to humans, hence social anxiety has been brought under control. During an online survey carried out in December 2020, 10% of respondents from the United States reported acquiring a new pet. This is an increase of 3% compared to May of the same year, when 7% of respondents acquired a new pet [4]. Furthermore, in Israel, Morgan et al. concluded that the interest of the dog adoption increased significantly in 2020, while dog abandonment did not change [3]. In contrast, in Poland, dog and/or cat adoptions remained constant or slightly increased, while significantly less quadrupeds were abandoned and surrendered to animal shelters. For instance, data from the inspection in the animal shelter in Gdynia, in northern Poland, indicates that in 2020, 693 dogs were abandoned and delivered to the animal shelter. This is a significant decrease compared with the number of abandoned dogs in 2019 (more than 1000 dogs were abandoned and delivered to the animal shelter in this year) [5]. This decreasing frequency is likely associated with the supportive effect of pets during the loneliness of lockdown and a positive impact on humans’ health and well-being.

When deciding to adopt and keep a pet, it is crucial to be aware that the low probability of SARS-CoV-2 transmission to a human does not mean such cases cannot occur. Data concerning its pathogenicity, routes of transmission, and strategies of elimination are still scarce. When analyzing the current literature indicating the direct or indirect correlation between animals and SARS-CoV-2 detection, as well as the frequent close contact between humans and animals, it is likely that, over time, not only human-to-animals transmission may occur, but also the reverse. Firstly, it is now widely accepted that wild fauna, most likely bats, constitute the primary reservoir of the SARS-CoV-2. Zhou et al. showed that the whole genomic sequence of bat coronavirus (bat-CoV) designated as RaTG13 is 96% identical to that of SARS-CoV-2 [6], suggesting that these viruses originate from bats. Furthermore, recent studies found that Malayan pangolins (*Manis javanica*) are frequently infected with CoVs. For instance, Li et al. indicated that pangolin-CoVs belonged to two different lineages. One lineage shared 97.4% amino acid identity to the receptor-binding domain (RBD) of the spike protein of SARS-CoV-2. Therefore, pangolins are considered to be a potential intermediate host for SARS-CoV-2 [7]. Secondly, the first reports of COVID-19 disease came from a “wet market”, named Huanan Seafood Market in Wuhan, China, where live, wild-caught animals and livestock are commonly sold. Due to these circumstances, it has been hypothesized that animals are involved in the spillover of SARS-CoV-2 [2]. Thirdly, it has been determined that SARS-CoV-2 belongs to the same family of viruses as SARS-CoV and Middle East respiratory syndrome coronavirus, designated as MERS-CoV. In the past, these viruses led to severe human diseases due to their transmission from animals to humans (SARS-CoV was associated with civet cats, while MERS-CoV was associated with dromedary camels) [8]. In 2002–2003, the first event occurred in China when SARS-CoV, which originated in bats, moved to humans, with palm civet cats being their intermediate host [9]. Recently, Sharma et al. evidenced the high phylogenetic similarity (approximately 79% similarity) of SARS-CoV-2 to SARS-CoV [10]. The second outbreak of human CoVs was caused by MERS-CoV and occurred in 2012 in Saudi Arabia. The virus was transmitted from bats to humans with dromedary camels acting as their intermediate host [11]. It was reported MERS-CoV infected about 2494 people, with 858 fatalities. Lastly, several studies suggest that captive animals such as farmed minks can directly infect humans with SARS-CoV-2, raising questions regarding their potential role in the virus circulation in the world [8,12,13]. Taken together, the SARS-CoV-2 outbreak may be fueled by zoonotic transmission from animals, including companion ones, to humans.

It is worth noting, that not only are the lives of human beings at risk, but there is an equal potential threat to the animal world via reverse human to animal transmission. Due to the fact that pets, especially dogs and cats, are in close contact with humans and inhabit the same environment, these animals are thought to be highly exposed to human pathogens including SARS-CoV-2 [12]. In this review, we reported cases of SARS-CoV-2 isolation from companion animals, such as cats, dogs, and ferrets, in order to identify potential zoonotic transmission from pets to humans, from pets to pets, and from humans to pets. Furthermore, in light of the fact that cats, mustelids (such as ferrets and minks), and hamsters are highly susceptible to COVID-19 infection under natural and/or experimental conditions, we made a juxtaposition of reports that use these animals as models to study pathogenicity, routes of transmission and development of vaccines and antiviral drugs. 

## 2. The Potential Risk of SARS-CoV-2 Transmission from Animals and the Role of ACE2 Receptors

SARS-CoV-2 uses angiotensin-converting enzyme 2 (ACE2) as a receptor for virus entry. Along with cell surface transmembrane serine protease 2 (TMPRSS2), viral fusion is carried out by the cleavage of receptor-binding domain (RBP) of viral spike protein. Sreenivasan et al. indicated that human ACE2 possess 24 amino acid residues that are required to bind the spike protein of SARS-CoV-2. Thus, any species that possesses similar amino acid residues would likely be susceptible hosts [14]. This discovery significantly increased the number of studies aiming to identify the structure of ACE2 in a variety of animal species. By comparing its similarity with human ACE2, it is possible to find animal models that contribute to the understanding of SARS-CoV-2 pathogenicity and routes of transmission. For instance, Luan et al. performed homology modelling of SARS-CoV-2 spike protein with ACE2 of several mammalian species, including companion animals. Based on the multi-scale computational approaches, authors predicted that SARS-CoV-2 could bind to ACE2 of ferrets, dogs, cats, hamsters, marmosets, and naked mole-rats, suggesting a broad host spectrum of the virus. The strength of association between the spike protein of SARS-CoV-2 and orthologues of ACE2 and TMPRSS2 was also analyzed in 215 vertebrate species [15]. Furthermore, Zhao et al. determined that dogs, cats, rabbits, and pangolins showed greater than 50% of hACE2 (human ACE2) binding, which indicates the high susceptibility of these species to SARS-CoV-2 [16]. The similarity of the ACE2 receptors with that of humans correlates relatively well with their infectivity. This may contribute to the risk of animal-to-human transmission of SARS-CoV-2 from animals having a high similarity ACE2 receptors to hACE2. It should be emphasized, however, that the there are many other factors that may favor or affect this transmission.

## 3. SARS-CoV-2 Detection in Cats and Their Zoonotic Potential

Cats are ranked as the second most common pets worldwide, followed by dogs. According to the 2020 American Veterinary Medical Association, in Canada alone, 48% of inhabitants own one or more cats, while 25% of US households own at least one cat [17]. Taking into account COVID-19 infections in pets under natural conditions, cats are more sensitive to SARS-CoV-2 than dogs [18]. Since the beginning of COVID-19 pandemic, 126 outbreaks of SARS-CoV-2 isolation from pet cats have been reported (67 outbreaks in the US and 59 outbreaks in other countries and areas; data: February 2021) [19].

The first report of SARS-CoV-2 isolation from pet cats comes from New York City, New York, the US. Two cats with mild respiratory illnesses were thought to have contracted the virus from people in their households or neighborhoods [20]. The reverse zoonotic transmission of SARS-CoV-2 was also observed in several independent studies in Hong Kong, China [21,22]. For instance, in March 2020, viral RNA was detected in the oral cavity, nasal, and rectal swab samples from a clinically healthy pet cat whose owner was infected with the virus [21]. In another study, 6 of 50 (12%) cats living with humans with SARS-CoV-2 infection were tested positive [22]. Also in China, this time in Wuhan—the epicenter of the COVID-19 pandemic, SARS-CoV-2 was detected from cats from three sources: animal shelters, pet hospitals, and COVID-19 patient families. A total of 102 serum samples were collected from these cats after the COVID-19 outbreak. SARS-CoV-2 was detected in 15 samples (14.7%). Among them, 11 (10.8%) had the viral neutralizing antibodies. This relatively high percentage of seropositivity in cats could be related to the large number of infected human cases having contact with pets [23]. Another case of human-to-cat transmission of SARS-CoV-2 was reported in Chile. On 5 May 2020, the cat-owners tested positive for SARS-CoV-2. Two days later, the male cat showed mild respiratory symptoms and tested positive. Four days after the male cat, the two female cats became positive, although asymptomatically. Additionally, one human and one cat showed antibodies against SARS-CoV-2 [24]. 

Including European countries, feline cases of COVID-19 have been reported in Belgium [25], Italy [26,27,28,29], the Netherlands [30], Spain [31,32], France [33,34], Germany [18], Switzerland [35], Russia [36], Croatia [37], Latvia [38], Greece [39] and the United Kingdom [40,41]. The first case of SARS-Co-V-2 isolation in pet cats in Europe was reported in Belgium. A cat kept at home with a COVID-19-infected owner became clinically ill, exhibiting respiratory problems accompanied by diarrhoea and vomiting. The specific viral sequence of SARS-CoV-2 was detected in the feaces and gastric vomitus of the cat; the sequence was identical to that of the cat owner, again suggesting human-to-cat transmission [25]. In other reports, COVID-19-infected cats were ill after direct contact with their owners, who were also tested COVID-19 positive. All these reports raised a huge public concern as the infected cats could play a role in SARS-CoV-2 transmission. To clear up any inconsistencies, Deng et al. collected 423 cat serum samples (including 48 samples in Wuhan and 42 samples in other 3 cities in China) for detection of the prevalence of SARS-CoV-2 specific antibodies. All samples were serologically negative to SARS-CoV-2 indicating that cats play a limited role in transmission during the COVID-19 pandemic [42]. This study is consistent with the work by Temmam et al., who also did not detect antibodies against SARS-CoV-2, nor viral RNA, in a cluster of 21 pets (9 cats and 12 dogs) kept in French households [43]. In conclusion, cats are highly susceptible to SARS-CoV-2, and may contract COVID-19 from pet owners. However, to date, there is no clear evidence that cat-to-human transmission of SARS-CoV-2 can occur. Table 1 shows the summarized information regarding reported COVID-19 on pet cats. 

## 4. SARS-CoV-2 Isolation in Dogs and Their Zoonotic Potential

According to the 2019–2020 American Pet Products Associations’, over 64 million North American inhabitants own at least one dog [3]. Including European countries, the number of pet dogs has seen a notable increase since 2010; the total amount of dogs reported in 2019 in Europe was estimated at 87.5 million. Germany ranked highest with a dog population of over 10 million in 2019, followed by the United Kingdom (UK) with 9 million [4]. As dogs are often in close contact with humans, especially during the COVID-19 pandemic, it is crucial to determine their susceptibility to SARS-CoV-2 and the impact of virus circulation globally. 

To date, there are scarce reports in which a pet dog was tested positive for COVID-19 disease. According to the World Organisation for Animal Health (OIE), 47 cases of SARS-CoV-2 have been reported in the US, while in other countries and areas the total notification was 37—as of February 2021 [19]). For instance, in the US (New York), SARS-CoV-2 was detected in housed pet dog (German shepherd) [50]. The dog showed signs of respiratory illness. The dog’s owner was also tested positive for COVID-19, suggesting reverse human-to-pet transmission. Furthermore, based on the Agriculture, Fisheries and Conservation Department (AFCD) report of Hong Kong, China, the virus was isolated from 2 of the total 15 dogs investigated (a 17-year-old Pomeranian and a 2.5-year-old German Shepherd). The viral RNA was detected in swabs in the nasal and oral cavities of tested dogs, and one of them developed specific antibodies against SARS-CoV-2. The virus titer was very low in the dog samples, and no clinical signs were observed [51]. Including European countries, SARS-CoV-2 has been isolated from pet dogs housed in the Netherlands [52], Italy [27], Croatia [39], Bosnia and Herzegovina [19], and Germany [53]. In the Netherlands, four domestic pets (one dog and three cats) were tested COVID-19 positive. The dog was suffering severe breathing problems and was euthanized due to the illness [52]. However, in Italy a research group of Patterson et al. reported a large-scale study to assess SARS-CoV-2 infection in 919 companion animals. Although no animals tested PCR positive, approximately 3.3% dogs had SARS-CoV-2 neutralizing antibody titers [27]. In both cases, dogs appeared to have contracted the virus from their COVID-19 owners, indicating human-to-pet transmission [54]. Several studies have attempted to isolate SARS-CoV-2 from pet dogs but most of them ended without obtaining COVID-19 positive samples. For instance, in France, neither RNA nor antibodies were detected in dogs living in the same room with veterinary students infected with SARS-CoV-2 [34]. Likewise, viral RNA was not detected in 12 dogs housed with confirmed infected individuals in Spain [32]. Taken together, these data suggest that dogs could catch the virus from people, but there was no sign that dogs were transmitting it in the reverse direction [55]. Table 2 shows the summarized information regarding reported COVID-19 on pet dogs.

## 5. Feline and Canine Models for SARS-CoV-2

As cats are susceptible to COVID-19 infection under natural conditions, these companion animals have been explored to study the SARS-CoV-2 pathogenesis and transmission. For instance, to answer the question, which biological mechanisms underlie the phenomenon of potential cat-to-cat transmission, Shi et al. found that juvenile cats were more vulnerable to SARS-CoV-2 infection than the older ones. This variation in viral dissemination in kittens may be reminiscent of the vast tissue tropism and indicate the variation in the disease severity exhibited by SARS-CoV-2 in humans. Viral RNA was detected in the nose and throat of both juvenile and sub-adult cats and caused inflammatory pathology in the respiratory tract of these animals which is consistent with severe COVID-19 in humans [51]. Cat-to-cat transmission under experimental conditions was also documented by Halfmann et al. [59] and Bosco-Lauth et al. [60]. In the first example, cats demonstrated SARS-CoV-2 shedding after 5 days postinfection. The virus was transmitted to naïve cats by close contact. SARS-CoV-2 was isolated from the tissues of upper respiratory tract such as the trachea, nasal turbinate, and esophagus [59]. In the second study, cats were highly susceptible to COVID-19 infection, with a prolonged period of oral and nasal viral shedding, and by direct contact transmitted the virus to other cats [60]. These experimental studies show that cats are not only susceptible to SARS-CoV-2 infection, but they also have a capability to transmit the virus to other cohoused cats. 

Cats are not often used as traditional animal models. However, the expression of ACE2 receptors, susceptibility of SARS-CoV-2 under natural conditions, and the ability to transmit the virus from cat to cat are factors encouraging their use as an alternative model to study SARS-CoV-2 pathogenesis [61]. As mentioned in earlier research of Bosco-Lauth et al., experimentally SARS-CoV-2-infected cats developed subclinical pathological changes in the upper respiratory tract early in the course of infection, with lower respiratory tract pathology later following viral clearance. This result suggests that viral infection of cats is not completely benign, and, hence, may be useful as an animal model to mild human COVID-19 infection [60]. Recently, Rudd et al. validated a feline model for SARS-CoV-2 infection; researchers found a significant correlation between the degree of clinical disease identified in infected cats (e.g., coughing, increased respiratory effort, lethargy, and fever) and pulmonary lesions observed due to SARS-CoV-2 infection, mimicking severe COVID-19 pathologies identified in hospitalized people [61]. Despite these promising preliminary results, the utility of cats as animal models for SARS-CoV-2 pathogenesis still requries further studies. 

Despite the high identity of canine ACE2 receptors with human ACE2 (83.4%), dogs are characterized by low susceptibility to SARS-CoV-2. The critical feature of this phenomenon is a single mutation (H34Y) in the canine ACE2 receptor that is not found in human or feline ACE2 [62]. Nevertheless, several reports show the viral infection of dogs under experimental conditions. For instance, three-month-old beagles were intranasally inoculated with an early pandemic strain (SARS-CoV-2/CTan/human/2020/Wuhan), and the viral RNA was detected in rectal swabs. Based on the sera collection and antibody detected by an ELISA, only two virus-inoculated dogs seroconverted. Neither antibodies nor virus has been detected in cohoused dogs [51]. These results are consistent with the study by Bosco-Lauth et al. Dogs did not shed the virus following infection [60]. These and other studies from the current literature lead to the conclusion that dogs are unlikely to be useful as animal models for SARS-CoV-2. 

## 6. Ferrets Models for SARS-CoV-2

To date, a natural SARS-CoV-2 infection has been recorded in pet ferrets housed in Spain [63] and Slovenia [64]. In Spain, Giner et al. evaluated the presence of SARS-CoV-2 antibodies in serum samples obtained from 127 household ferrets (*Mustela putorius furo*). Two ferrets tested positive (1.57%). SARS-CoV-2 antibodies persisted at detectable levels in a seropositive SARS-CoV-2 domestic ferret beyond 129 days since the first—time antibodies were detected [63]. The first detection of SARS-CoV-2 from pet ferrets was reported in Slovenia in late December 2020. The ferret lived in the home with a person suffering from COVID-19 and exhibited gastrointestinal signs [64]. 

Ferrets are common laboratory models for SARS-CoV-2 due to their high susceptibility to the virus and the ability to transmit it to other ferrets. Independently, Kim et al. [65], Yoo et al. [54], Schlootau et al. [66], Shi et al. [51], and Richard et al. [67] determined that the virus was successfully transmitted to cohoused ferrets (direct contacts) and via the airborne route (indirect contacts). For instance, in a study by Kim et al., experimentally infected ferrets displayed either no clinical symptoms or exhibit elevated body temperature and loss of appetite. Viral RNA was detected in nasal lavages after 2, 4, 6, and 8 dpi (days post-infection). This result, together with the data obtained from other studies regarding the SARS-CoV-2 infection response in ferrets after injection of different viral loads, demonstrated that the virus could replicate in the upper respiratory tract of ferrets, showing a disease pattern similar to that of humans [65]. In addition, unlike mice and rats, ferrets exhibit the cough reflex; as coughing is the most frequently reported symptom in cases of SARS-CoV-2 infection, these animals represent promising models for this virus [68].

Moreover, ferrets have proven to be an appropriate animal model for testing safety and efficacy of SARS-CoV-2 vaccines and antiviral drugs [69]. Park et al. evaluated the antiviral efficacies of three FDA-approved drug candidates against SARS-CoV-2, including lopinavir-ritonavir, hydroxychloroquine sulfate, and emtricitabine-tenofovir. The virus titers in nasal washes, stool specimens and respiratory tissues were similar between all groups treated with drug candidates. Only the emtricitabine-tenofovir-treated group of ferrets showed lower virus titers in nasal washes at 8 days postinfection compared to the PBS-treated control group. Thus, although all antiviral drugs marginally reduced the overall clinical scores of infected ferrets, they did not significantly affect in vivo virus titers [70]. Furthermore, de Vries et al. designed lipopeptide fusion inhibitors that block fusion of the viral and host cell membranes during first step of SARS-CoV-2 infection. Daily intranasal administration of these inhibitors to ferrets completely prevented SARS-CoV-2 direct-contact transmission during 24-h cohousing with infected animals. This study reveals the high potential of lipopetides for effective intranasal prophylaxis to reduce transmission of SARS-CoV-2 [71].

Recently, Blanco-Melo et al. [72] and Liu et al. [73] described antiviral gene signatures induced in ferret models after SARS-CoV-2 infection that can potentially act as antiviral drug targets. Furthermore, Cox et al. demonstrated that treatment of infected ferrets with twice-daily MK-4482/EIDD-2801 significantly reduced upper respiratory tract SARS-CoV-2 load and completely suppressed spread to untreated contact animals. This drug is an analog inhibitor of influenza viruses, EIDD-2801 (or MK-4482), being repurposed against SARS-CoV-2 and is currently in phase II/III clinical trials [74]. These data taken together indicate that ferrets seem to be suitable animal models for studying the pathogenesis of SARS-CoV-2 and COVID-19 drug and vaccine development.

## 7. SARS-CoV-2 Isolation in Farmed Minks and Their Zoonotic Potential

Not only domestic, but also farmed and free-living animals are assumed to be involved in SARS-CoV-2 circulation in the world. SARS-CoV-2 has been detected in farmed minks (*Neovison vison*) in several European countries including Poland [75,76,77], the Netherlands [78], Spain [79], Denmark [80], Italy [81], Greece [82], Sweden [83], Latvia [84], and Lithuania [85]. In addition, infected minks have been observed and reported in the US and Canada [86]. To date (i.e., February 2021), SARS-CoV-2 was detected in 409 mink farms worldwide (mostly in Denmark (290 farms) and the Netherlands (69 farms)) [19]. First reports of SARS-CoV-2 isolation from two minks were documented in the Netherlands (April 2020). Viral RNA was detected in the airborne inhalable dust on the mink farms, suggesting that dust and/or droplets are means of transmission between the minks and indicating possible exposure for the workers on the farms [78]. In turn, in the US, 17 cases of SARS-CoV-2 were identified in farmed minks, most of which were located in Utah, with one farm in Wisconsin, and another in Oregon (November 2020). In one case, the probable transmission of mink-to-human was observed in a person in close contact with the animals on the farm. Again, this observation indicates the potential threat of mink-to-human transmission of the virus [86]. Therefore, due to the increasing SARS-CoV-2 occurrence in people connected to mink farms, in many countries, preventive elimination of these animals has been implemented (some farms have suspended or definitively terminated their activities). For instance, in Poland, the number of active farms had decreased from 354 in 2020 to 266 at the beginning of 2021. From 28 farms located in different regions of Poland, SARS-CoV-2 was detected in one of them (Pomorskie voivodeship (in the north of Poland)). In this study, 50 throat swabs and 150 serum samples from healthy bred minks were tested. SARS-CoV-2 was isolated from 35 of the 50 throat swabs (70%) and 45 of the 150 serum samples (30%) [75]. In the next stage of SARS-CoV monitoring of this mink farm, throat swabs were collected from 91 minks, and in 15 samples, SARS-CoV-2 was detected (16.5%) [76]. All these reports taken together show that, considering captive animals, minks possess a high zoonotic potential of SARS-CoV-2 transmission to humans. 

## 8. Hamsters Models for SARS-CoV-2

Considering SARS-CoV-2 infections, there is no report of its detection in hamsters under domestic conditions (hamsters as pets). However, these animals are suitable experimental models that provide new aspects of viral pathogenesis and transmissibility as well as successful treatment. Among other studied animal models, the alignment of the ACE2 proteins of humans and hamsters is one of the highest [87]. Thus, the spike protein of SARS-CoV-2 may interact more efficiently with hamster ACE2 than ACE2 from other animal models. In one study, hamsters were intranasally inoculated with SARS-CoV-2, and viral antigens were detected, mainly in the lungs. Notably, SARS-CoV-2 was transmitted efficiently from inoculated hamsters to cohoused naïve hamsters by direct contact and via aerosols. All hamsters recovered, and neutralizing antibodies were detected within 14 dpi [88]. This study was consistent with a study by Boudewijns et al., which aimed to test the mRNA of the hamsters’ cytokine profiles. The authors observed induction of interferon-γ and pro-inflammatory chemokines/cytokines upon SARS-CoV-2 infection. Primarily inoculated animals developed clinical signs, including lethargy, ruffled fur, hunched back posture, tachypnea, and loss of bodyweight. [89]. Similar results related to viral susceptibility and transmissibility have been reported by Zhang et al. [90] and Lau et al. [91].

The hamster model was also used in studies regarding the effect of age and sex in the severity of the COVID-19 disease in humans. For instance, Osterrieder et al. followed the course of SARS-CoV-2 infection in young and aged Syrian hamsters. Although viral replication in the upper and lower respiratory tract occurred regardless of the animals’ age, hamsters infected at older ages experienced a more pronounced weight loss than younger animals. Histopathological analysis showed a critical age-dependent influx of immune cells into the lungs, which happened earlier and stronger in young animals [92]. These results observed in hamster models appear to mirror age-dependent differences in human patients.

Lastly, hamsters have proven to be suitable for evaluating vaccines [93,94,95] and antiviral drugs [96,97] against SARS-CoV-2. With the use of Syrian hamster models, Yuan et al. identified ranitidine bismuth citrate, a commonly used drug for the treatment of *Helicobacter pylori* infection, as a potent anti-SARS-CoV-2 agent, both in vitro and in vivo. The drug suppressed SARS-CoV-2 replication, leading to decreased viral loads in both upper and lower respiratory tracts and relieved virus-associated pneumonia in tested models [96]. Furthermore, Kaptein et al. determined that a potential candidate of SARS-CoV-2 treatment—favipiravir—significantly reduced infectious virus titers in the lungs of hamsters and markedly improved lung histopathology. This result indicates that favipiravir exhibits a marked protective effect against SARS-CoV-2 in the hamster model [97]. 

The current dynamics of SARS-CoV-2 affecting humans during the COVID-19 pandemic necessitates further detailed investigations concerning the transmission ability of the virus from humans to animals and vice versa. A summary of the data gathered from all studies included in this review is shown in Figure 1. Although in this review several reports regarding SARS-CoV-2 detection in hamsters under experimental infection have been presented, to date, there is no case of natural infection in these animals. Thus hamsters were excluded from this figure.

## 9. The Need for the Analysis of Epizootiological and Social Risks Analysis in the SARS-CoV-2 Transmission

One of the critical objectives of the COVID-19 pandemic is thought to be the development of procedures among experts in the field of infectious diseases and ecology to reduce the risk of SARS-CoV-2 transmission in their natural environment. Undoubtedly, the most crucial point of epidemiological proceedings at present should be to limit the transmission of the virus between humans. However, certain social and animal habits and ecological interactions in the environment cannot be ignored. Table 3 summarizes the factors that reduce or increase the risk of transmission of SARS CoV-2 from animals to humans [98,99,100,101,102,103]. 

## 10. Perspectives

There is a steady increase in the number of reports on the detection of COVID-19 in companion animals around the world. Further studies are required to evaluate the potential of pets to serve as efficient reservoir hosts that can further alter the dynamics of human-to-human transmission. Research collaboration between epidemiologists and veterinary and wildlife specialists is becoming a necessity to evaluate and identify the possible risk factors of transmission between animals and humans. Such cooperation will help to devise efficient strategies for the management of emerging zoonotic diseases. In contrast, we must take care with publicizing the results of studies concerning SARS-CoV-2 isolation from pets in non-scientific communication channels. The impacts of presenting that information to an audience that is not familiar with the scientific community can cause unnecessary panic and catalyze severe consequences for animals and public health [104].

As people and animals can both affected by SARS-CoV-2, it is highly recommended to limit contact with animals if possible. It should be initiated to farmed, companion, free-living and wildlife animals. If people have to look after their pets, they should maintain good hygiene practices. Animals belonging to owners infected with SARS-CoV-2 should be kept indoors in line with similar lockdown recommendations for humans applicable in the country or area.

## 11. Conclusions

The reports introduced in this review refer to cases of SARS-CoV-2 isolation from pets and statistically inform that, to date, there are no signs that these animals, especially dogs and cats, are a source of infections for humans. The evidence only leads us to cases in which human beings (guardians or handlers) infected by COVID-19 transmitted the virus to their pets or charges, cases in which the virus was experimentally inoculated. Given the magnitude of COVID-19 in humans, the lack of any case documenting COVID-19 being transmitted from pets to humans should provide the necessary comfort that our feline and canine friends are not virus propagation factors for humans. However, it is worth noting that several factors, including illegal trade of wildlife animals or the deep anthropization of the territories inhabited by wildlife animals, may have a huge impact on environmental changes, with the potential consequences linked to the appearance of SARS-CoV-2 in humans and their pets due to its transmission from wildlife animals. 

## Figures and Tables

**Figure 1 viruses-13-01149-f001:**
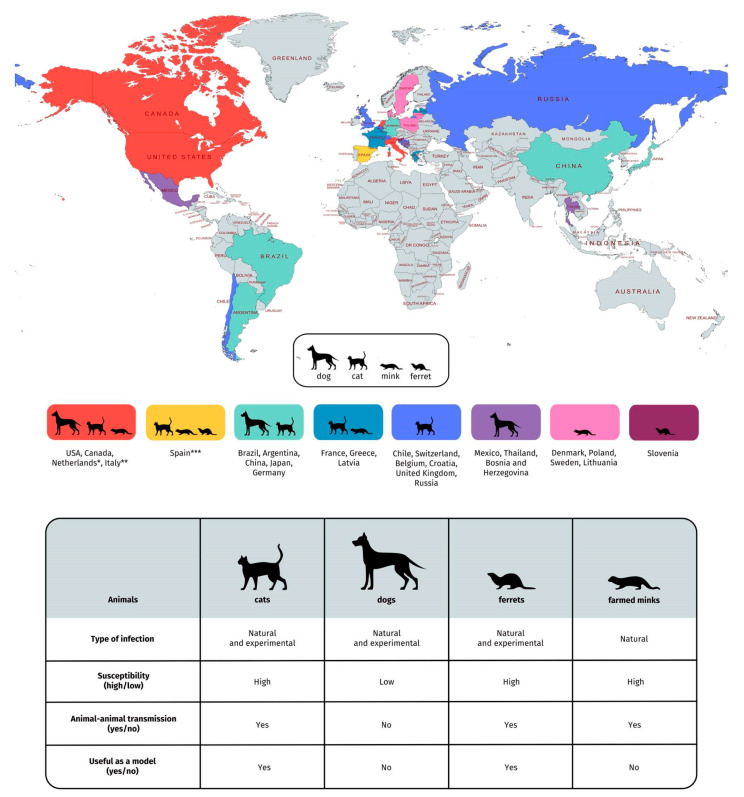
Summary of findings of the SARS-CoV-2 infection in companion animals (including cats, dogs, and ferrets) and farmed minks and reports of SARS-CoV-2 natural infection in these animals (25 May 2021) [19]. * Data of dog and cat isolation in the Netherlands are collected from [30,52]; ** Data of dogs isolation in Italy are collected from [27]; *** Data of pet ferrets isolation in Spain are collected from [63]. We certify that we stays neutral with regard to jurisdictional claims in this map.

**Table 1 viruses-13-01149-t001:** Summarized reports of COVID-19 infection in cats kept as pets (from the highest number of tested cats in particular study to the lowest) *.

Number of Tested Cats	COVID-19 Tested Positive Cats	Country/Area	Date (Month and Year of Publication)	Reference
920	6	Germany	December 2020	[18]
191	11	Italy	December 2020	[26]
50	6	Hong Kong, China	December 2020	[22]
22	1	France	June 2020	[34]
11	3	The Netherlands	May 2020	[30]
8	1	Spain	August 2020	[32]
5	5	Japan	January 2021	[44]
4	10	Brazil	April 2021	[45]
4	3	United Kingdom	July 2020	[41]
4	3	Chile	March 2021	[24]
3	3	Latvia	February 2021	[38]
2	2	The Switzerland	March 2021	[35]
2	2	Canada	January 2021	[46]
2	2	Argentina	November 2020	[47]
1	1	Belguim	April 2020	[25]
1	1	Hong Kong, China	March 2020	[48]
1	1	Spain	May 2020	[31]
1	1	Brazil	October 2020	[49]
1	1	France	May 2020	[33]
1	1	Italy	February 2021	[27]
1	1	Italy	September 2020	[28]
1	1	Greece	December 2020	[41]
1	1	Croatia	December 2020	[39]
1	1	Italy	March 2021	[29]
1	1	Russia	June 2020	[38]

* due to the relatively higher number of naturally acquired SARS-CoV-2 infection in domestic cats in the US (67 studies—February 2021) than in other countries and areas [19], these reports were not included in this table.

**Table 2 viruses-13-01149-t002:** Summarized reports of COVID-19 infections in dogs kept as pets (from the highest number of tested cats in particular study to the lowest one) *.

Number of Tested Dogs	COVID-19 Tested Positive Dogs	Country/Area	Date (Month and Year of Publication)	Reference
451	15 *	Italy	December 2020	[27]
18	10	Mexico	December 2020	[56]
15	2	Hong Kong, China	March 2020	[51]
9	29	Brazil	April 2021	[45]
4	4	Argentina	November 2020	[47]
4	4	Japan	August 2020	[44]
3	3	Croatia	April 2021	[39]
2	2	Germany	November 2020	[53]
1	1	The Netherlands	May 2020	[52]
1	2	Canada	October 2020	[57]
1	1	Bosnia and Herzegovina	February 2021	[19]
1	1	Thailand	May 2021	[58]

* due to the relatively higher number of naturally acquired SARS-CoV-2 infection in domestic dogs in the US (47 studies—February 2021) than in other countries and areas [19], these reports were not included in this table.

**Table 3 viruses-13-01149-t003:** Factors influencing the potential risk of transmission of SARS CoV-2 from animals to humans.

The Potential Risk of SARS CoV-2 Transmission	
Diminishing Factors	Potentiating Factors
Use of restrictive personal protection measures	Social habits (close contact with wild animals, e.g., traditional cuisine or natural medicine)
Suitable procedures for the transport and handling of diagnostic samples	Lack of testing procedures and security measures in the transport of samples
Non-invasive methods of collecting samples from animals that can reduce the risk of e.g., bites	Low social and living standards
Information campaigns among the population about the possible threat and proper conduct of animals	Possibility of contact of domestic animals with wild ones
Establishing and respecting the law on the circulation and trade of animals potentially acting as pathogens’ carriers	Illegal trade of wild animals
Reduction in amateur, wildlife tourism and canyoning	The phenomenon of urbanization and globalization (appearance of wild animals in cities and occupation of new wild territories by humans)

## Data Availability

The authors confirm that the data supporting the finding of this study are available within the article.

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
