# Peer review of "Current State of Knowledge about Role of Pets in Zoonotic Transmission of SARS-CoV-2"

_viruses, 2021, doi:10.3390/v13061149_

Round 1

Reviewer 1 Report

Current state of knowledge about role of pets in zoonotic trans-mission of SARS-COV-2

Mateusz Dróżdż et al.

This manuscript is a comprehensive and well written review of the potential role of pet species in transmission of COV-2 and their use as experimental models of disease. Aside from some stylistic suggestions the submitted MS is highly relevant and publishable.

Specific suggestions for the Authors. Please take most of these as suggestions for readability.

LN15 “arising” suggest ‘increasing or emerging’

LN18 “od” misspelled ‘of’ “aroses” misspelled ‘arouses’

LN22-27 suggest present tense “presented” suggest ‘present’, “documented” suggest ‘document’, “suggested” suggest ‘suggest ‘

LN32 “keep” suggest ‘observe’

LN38 “emotiona!’ misspelled ‘emotional’

LN41 “Nevertheless, over time, after reporting data evidencing very low probability of transmitting” suggest ‘reports indicated low probability of transmission’

LN46 “got” suggest ‘acquired’

LN48 remove “incidences”

LN51 “approved” suggest ‘delivered or donated’ also in LN53

LN60-61 “However, over time, zoonotic potential of the virus is assumed to be increased,

leading, in consequence, to animal-to-human transmission.” This is a key sentence as it introduces an important concept described in detail in the rests of the paragraph. This sentence is confusing. Does it refer to increased recognition that COV-2 likely came from a bat source perhaps via another animal/

LN90-91 “It is worth to note, that not only are the lives of human beings at risk, but there is an

equal potential threat to the animal world.” This is an important concept introducing the paragraph.

 Suggest ‘It is worth to note, that Not only are the lives of human beings at risk, but there is an

equal potential threat to the animal world via reverse human to animal transmission.’

LN110-111 “it is possible to find animal model that is required to understand the pathogenicity…” suggest ‘that contributes to the understanding…’

LN133 “New York, the US” suggest ‘New York, New York, US’

LN164 “for detecting” suggest ‘for detection of …’

LN187 “there were” suggest ‘there are…’

LN207-208 “Several studies have attempted the SARS-CoV-2 isolation from

pet dogs without indicating COVID-19 positive samples.”  Suggest ‘have attempted to isolate SARS…’ and “without indicating” does this mean without successfully isolating virus?

LN241 “encouraging them” suggest ‘validating their use or encouraging their use…’

LN259 “with Chinese virus…” this phrase has unfortunately been used by some as a political weapon. Suggest ‘inoculated with an early pandemic strain (SARS … Wuhan).’ It will be obvious to the reader where the strain came from.

LN267-268 and else where “For instance, …” this way of introducing evidence gets repetitive (LN291-292, LN373-374). In this case one could simply change this to ‘ Giner et al. evaluated the presence of SARS-CoV-2 antibodies in serum samples obtained from 127 Spanish households with pet ferrets (Mustela 269 putorius furo).’

LN273 OIE defined earlier

LN317 “Recently” this is no longer so recent suggest ‘The virus…’

LN319 “notified” suggest ‘reported or noted’

LN321-322 “farmed minks” the sentence starts by referring to mink farms change these refs to “farmed minks’ to simply ‘farms’

LN323 “Nerherlands’ misspelled ‘Netherlands’

LN334 “In this country…” suggest ‘In Poland…’

LN340 (15/91, 16.5%) the numerator and denominator are already in the sentence suggest just (16.5%)

LN342 “farmworkes” misspelled ‘farm workers’

LN359 “Primarily inoculated…” confusing suggest ‘Just inoculated…’

LN360-362 “Other studies related to viral susceptibility and transmissibility come from (as the example) Zhang et al. [85] andLau et al. [86].  Are the results of these studies important enough to mention the results?  If so suggest “Similar results have been reported by …’

Figure Very nice summary figure.

LN398 “Nowadays…” suggest starting with ‘One of the …’

LN399 “risk of infections from animals in their natural environment” suggest ‘risk of COV-2 transmisson in their…’

LN410-411 “animal health sectors….” Suggest ‘veterinary and wildlife specialists…’

LN425 “The introduced in this review reports refer” suggest ‘The reports introduced in this review…’

LN427-428 “us’ and “beings” delete

References:

assessed should be ‘day month year’ throughout

space follows year, volume, pg throughout

some references lack “accessed…” such as LN 456, LN465, LN467

“accessed” is sometimes misspelled ie. LN 461, LN499

Journals need to be abbreviated  See LN703

Journal abbreviations need a period after each abbreviation ‘J. Clin. Microbiol.’

When DOI address is given “doi” should be in caps

Author Response

Dear Reviewer,

Thank you for providing insightful feedback on ways to strengthen our paper titled 'Current state of knowledge about role of pets in zoonotic transmission of SARS-COV-2. We have incorporated changes that reflect the detailed suggestions you have graciously provided. We also hope that our edits and the responses we provide below satisfactorily address all the issues and concerns you have noted.

Sincerely, 

Mateusz Drozdz

Reviewer 2 Report

The Authors described the current status about the role of pets in zoonotic transmission of SARS-CoV-2. The paper is clear, well-written and may be published on "Viruses" journal.

Just very few minor points:

  • line 32: I prefer the term "physical" instead "social", because humans are social animals, then the distancing  may be physical only.
  • table 3: I think that in the "Potentiating factors" column the "occupation of new wild territories by humans" may be useful to add. The "appareance of wild animals in cities" alone does not explain the possibilities to meet new potential pathogens.
  • Conclusion: I think that the Authors have to add few lines about the deep anthropization of the territories may lead to changes in the environment (i.e. climate changes) with the consequente links to the appearance of new potential pathogens for humans and their pets.

Author Response

Dear Reviewer, 

Thank you for providing insightful feedback on ways to strengthen our paper. We have incorporated changes that reflect the detailed suggestions you have graciously provided. We also hope that our edits and the responses we provide below satisfactorily address all the issues and concerns you have noted.

Sincerely, 

Mateusz Dróżdż
